# Investigating the Biocontrol Potential of the Natural Microbiota of the Apple Blossom

**DOI:** 10.3390/microorganisms10122480

**Published:** 2022-12-15

**Authors:** Anya Schnyder, Leo Eberl, Kirsty Agnoli

**Affiliations:** 1Institut für Veterinärbakteriologie, Universität Bern, 3001 Bern, Switzerland; 2Department of Microbiology, Institute of Plant and Microbial Biology, University of Zürich, 8008 Zurich, Switzerland

**Keywords:** biocontrol, fire blight, *Erwinia amylovora*, apple blossom bacteria, antifungal activity, anti-oomycete activity

## Abstract

*Erwinia amylovora*, the causative agent of fire blight, leads to important economic losses of apple and pear crops worldwide. This study aimed to investigate the potential of the resident microbiota of the apple blossom in combatting plant disease-causing organisms, with a focus on controlling fire blight. We obtained 538 isolates from sites around Canton Zurich, which we tested for activity against *Pectobacterium carotovorum* and *E. amylovora.* We also evaluated the isolates’ activity against oomycete and fungal pathogens. Nine isolates showed activity against *P. carotovorum*, and eight of these against *E. amylovora*. Furthermore, 117 showed antifungal, and 161 anti-oomycete, activity. We assigned genera and in some cases species to 238 of the isolates by sequencing their 16S RNA-encoding gene. Five strains showed activity against all pathogens and were tested in a detached apple model for anti-*E. amylovora* activity. Of these five strains, two were able to antagonize *E. amylovora*, namely *Bacillus velezensis* #124 and *Pantoea agglomerans* #378. We sequenced the *P. agglomerans* #378 genome and analyzed it for secondary metabolite clusters using antiSMASH, revealing the presence of a putative bacteriocin cluster. We also showed that *B. velezensis* #124 exhibits strong activity against three different fungi and two oomycetes in vitro, suggesting a broader capacity for biocontrol. Our results showcase the protective potential of the natural apple blossom microbiota. We isolated two candidate biocontrol strains from apple blossoms, suggesting that they might persist at the most common entry point for the causative agent of fire blight. Furthermore, they are probably already part of the human diet, suggesting they might be safe for consumption, and thus are promising candidates for biocontrol applications.

## 1. Introduction

The strong market and technological competition among crop-producing countries has forced the agricultural industry to increase yields while reducing production costs [1]. This has led to a dramatic increase in the use of pesticides to control phytopathogens; however, in recent years tighter legislation has reduced the number of conventional pesticides that meet current safety standards, and the EU sustainable pesticide use directive requires that crop protection should avoid conventional chemical application where possible [2]. This is driving the search for more environmentally friendly methods to control plant disease. In contrast to chemical pesticides, biological approaches using microbial antagonists are considered an environmentally friendly and more sustainable option in pest control [3,4]. 

One major agricultural disease requiring new prevention options is fire blight. Fire blight is a globally occurring disease of *Rosaceae* trees and causes severe economic shortfalls due to loss of apple and pear crops, estimated at over USD 100 million in the USA annually [5]. Infected plants bear spoiled, unsaleable fruit, and severe infections can spread to the roots and lead to death of the tree [6]. Fire blight is caused by the bacterium *E. amylovora*, which primarily enters through the stigma of blossoms. *E. amylovora* infections can persist over the winter, to re-emerge in the spring. Under humid conditions, infected trees ooze bacterial exudate, which acts as an inoculum for further spread of the disease [7]. As a result, there is a heavy risk of severe outbreaks when the spring is mild and humid. Such conditions are becoming more prevalent with the progression of global warming [7]. Fire blight presents a particular problem due to lack of effective control measures. Current methods include aggressive pruning, copper sprays, and antibiotic application [8,9]. The application of streptomycin is the most effective treatment, but is controversial due to the development of antibiotic resistance, and is no longer approved for use in the EU. As a result, alternative strategies to protect apple and pear trees are urgently required. There are various reports of bacteria capable of fire blight biocontrol [10,11,12,13]. For the best results, these must be applied to blossoms shortly prior to challenge by *E. amylovora,* and multiple applications of bacterial biocontrol agents are recommended, with at least two applications made between 25% and 90% bloom [9,14]. However, all strains that are currently approved in the EU are from *Bacillus amyloliquefaciens* and *Bacillus velezensis*, for example MBI600 (Serifel), QST713 (Serenade) and D747 (Double Nickel) (https://ec.europa.eu/food/plant/pesticides/eu-pesticides-database/active-substances/, (accessed on 10 November 2022). A recent study of the apple blossom microbiome found a high abundance of natural *E. amylovora* antagonists in apple blossoms [15], and the work presented here aimed to isolate strains from the same habitat that have the potential to counteract *E. amylovora*.

This study aimed to investigate the potential of the resident microbiota of the apple blossom in combatting plant disease-causing organisms, especially fire blight. We obtained 538 isolates, which we tested for activity against *Pectobacterium carotovorum* and *E. amylovora,* and furthermore against oomycete and fungal pathogens. We found two isolates that showed activity against all tested pathogens in vitro and that were able to antagonize *E. amylovora* in an apple model of infection.

## 2. Materials and Methods

### 2.1. Isolation of Strains from Apple Blossom 

Apple blossoms were collected in spring 2021 from four different sites around Zürich (Wädenswil, Rüti, Feldbach in Hombrechtikon, and Wülflingen in Winterthur) in four different phenological developmental states, at BBCH index 57 (pink bud stage), 63 (30% flowers open), 65 (full flowering), and 67 (flowers fading). The apple tree types from which the blossoms were collected were Topaz, Red Topaz, Remo, Odysso, Braeburn and Gravensteiner, and had not been treated with conventional pesticides or biocontrol agents. Additionally, a mixed sample from two gardens in Rorbas, ZH was included. In total, 25 samples were taken, each of which consisted of approximately 100 blossoms. Each sample was homogenized in 20 mL 0.9% NaCl using an immersion blender and filtered using a BioReba extraction bag (Bioreba AG, Reinach, Switzerland). Three media were used for isolation: NSLB, M9 glycerol and PCAT (see below), and the remainder of each sample was frozen as a glycerol stock. Plates were kept at room temperature for colonies to develop. Colonies with bacterial appearance were restreaked on the same medium to gain single colonies. Colony-purified isolates were stored as glycerol stocks.

### 2.2. Strains and Media

Strains and primers used in this study are listed in Table 1 and Table 2, respectively. Apple blossom isolates are detailed in Appendix A. Unless otherwise stated, strains were grown aerobically in no-salt LB broth at 30 °C. Media used for isolation of microorganisms from apple blossoms were no-salt LB (NSLB) (1% tryptone *w/v* (VWR, Radnor, Pennsylvania, USA), 0.5% yeast extract *w/v* (Thermo Fisher Scientific, Reinach, Switzerland), solidified with 1.5% *w/v* agar), M9 minimal salts with 1% *w/v* glycerol as the carbon source + 0.1% *w/v* casamino acids (CAA) (Thermo Fischer Scientific, Waltham, MA, USA) [16], and PCAT [17]. Purified isolates were grown overnight in NSLB broth (30 °C with shaking). Anti-fungal assays were carried out on malt extract agar (Thermo Fischer Scientific, Waltham, MA, USA) supplemented with additional agar to a total concentration of 2% *w/v*. Anti-*Phytophthora* assays were carried out on pea agar with 1% glucose (the content of one 420 g tin of peas was rinsed with tap water and simmered with a cover to limit evaporation in 1.5 L distilled water for 1 h. The liquid was separated from the peas by filtration through a piece of paper towel, and 1% *w/v* glucose and 1.5% *w/v* agar (Condalab, Madrid, Spain) were added to the filtrate before autoclaving). Two fungi were isolated from apples as test microorganisms by placing a piece of infected apple tissue in the middle of a malt extract agar plate and incubating at room temperature in the dark. Two samples showing rapid growth were chosen and sequence-typed using ITS primers (see below).

### 2.3. Antimicrobial Screening

Isolates were screened for activity against three types of microorganisms: bacteria, represented by *P. carotovorum* and *E. amylovora*; fungi, represented by *Fusarium solani*; and oomycetes, represented by *P. cinnamomi* and *P. infestans.* Two additional fungi were also tested against selected isolates. These were *Fusarium graminearum* and *Diplodia seriata.*

#### 2.3.1. Antibacterial Assays

M9 glycerol agar + CAA (30 mL) was poured into each 12 cm square plate. Wooden culture sticks were dipped into overnight cultures of the test isolates and used to inoculate the plates (9 strains per plate). Plates were incubated at 30 °C for 24 h before carefully pouring on a second layer of M9 glycerol agar + CAA (25 mL) containing 50 µL of an overnight culture of *P. carotovorum* or *E. amylovora.* Of critical importance here was the temperature of the agar to which the *P. carotovorum* or *E. amylovora* was added (cool enough to touch, so as not to kill the bacteria, but warm enough for easy pouring, ~45 °C). Plates were incubated at 30 °C. Isolates with activity against the test bacterium produced a growth-free halo in the second agar layer.

For confirmation of positive results, the assay was repeated in triplicate using standardized test cultures at OD_600_ 2. Ten μL of each test isolate was pipetted into holes punched in the bottom agar layer using the wide end of a 10 μL pipette tip.

For the initial screening, activity was scored relative to the strongest observed activity across the group. The zone of inhibition (ZOI) was measured by putting a ruler through the middle of the test colony and measuring the distance from the edge of the colony to the *Erwinia* lawn. The scale used was as follows: 0, no activity; 1, some activity, giving a diffuse halo that could not be accurately measured; 2, defined halo ≤ 5 mm, 3, halo > 5 mm.

#### 2.3.2. Anti-Phytophthora and Antifungal Assays

A dual culture technique was used for screening the apple blossom isolates for activity against oomycetes and fungi. Overnight cultures of the isolates were grown, and 20 µL aliquots spotted at three points around the edge of a pea agar 1% glucose plate (for the oomycete assays) or malt extract agar (for antifungal assays). The isolates were allowed to grow at 30 °C for 24 h, before a plug of oomycete/fungus (taken from the leading edge of a growing plate) was added to the center of the Petri dish. Plates were sealed with parafilm and kept at room temperature in the dark until the oomycete/fungus on the control plate (without test isolates) reached the outer edge of the plate. In the initial screen, the size of the ZOI was measured by putting a ruler through the middle of the test colony and the plug and measuring the distance from the edge of the colony to the edge of the fungal/oomycete growth. 

Antifungal and anti-*Phytophthora* activity was scored as follows: 0, no inhibition; 1, some inhibition observed. In some cases, this was in the lee of each colony (not directly between the leading edges of the colony and fungus/oomycete). ZOIs > 0 and <5 mm were also scored as 1. A score of 2 was given where ZOI ≥ 5 mm and <10 mm; 3, ZOI ≥ 10 mm and <15 mm; 4, ZOI ≥ 15 mm.

### 2.4. 16S rRNA Sequencing of Bacterial Strains

16S rRNA sequencing of apple blossom isolates was performed with primers PRK349F and PRK806R [21] using Terra PCR Direct Polymerase Mix (TaKaRa Bio, Shiga, Japan) following the manufacturer’s instructions. Colony material was added to PCR tubes directly as template. The annealing temperature used was 55 °C, and the extension time was 40 s. The expected fragment size was ~450 bp. Samples were purified and sequenced by Microsynth (Microsynth AG, Balgach, Switzerland), using the amplification primers. Bacterial genus was determined by NCBI BLAST search against the non-redundant DNA database (nr). For more accurate taxonomic resolution, the procedure was repeated with primers UP-1 and UP-2r [22,23] giving a 1200 bp *gyrB* amplicon.

### 2.5. ITS Sequencing of Fungi

During isolation of strains from apple blossoms, several fungi were isolated. These were sequence-typed using ITS4 and ITS5 primers, which amplify a 672 bp fragment of the internal transcribed spacer (ITS) between the 18SrRNA and 23SrRNA [24]. Colony material was resuspended in 30 µL 0.2% sodium dodecyl sulfate (SDS) solution and heated for 4 min at 90 °C. Promega Go Taq was used for amplification (Promega, Wisconsin, USA), and the PCR reaction was set up following the manufacturer’s instructions, with the exception that 1 μM final concentration of each primer was used in the reactions. The annealing temperature used was 55 °C, and extension time was 40 s. Samples were purified and sequenced by Microsynth, using the amplification primers. The genus or species of each fungus was determined by NCBI BLAST search against the non-redundant DNA database (nr).

### 2.6. Genomic DNA Preparation for Sequencing

The strain *P. agglomerans* #378 genome was sequenced at the Functional Genomics Center Zürich, Switzerland with PacBio Sequel technology. DNA was extracted with DNAElute Bacterial Genomic DNA Kit NA2110-1KT (Sigma-Aldrich, St. Louis, MO, USA) according to the manufacturer’s protocol and checked for concentration and purity by NanoDrop (Thermo Fischer Scientific, Waltham, MA, USA) and Qubit (Thermo Fischer Scientific, Waltham, MA, USA).

### 2.7. Apple Model for Testing of Biocontrol Potential against E. amylovora ex Vivo

Isolates showing activity against *E. amylovora* antagonists in vitro were further tested using an apple model modified from Klee et al., 2019 [25]. 

Inoculant preparation: Strains to be tested were grown overnight in NSLB at room temperature. Cultures were adjusted to OD_600_ 1 for single strain suspensions and OD_600_ 2 where the test strains were to be mixed with *E. amylovora* (to give OD_600_ 1 after 1:1 mixing). MilliQ water was used as a control. The CFU present in each inoculant was enumerated by plating serial dilutions on NSLB plates. Ten μL suspension at OD_600_ 1 corresponded to ~3 × 10^6^ CFU for #378, ~400,000 CFU for #124 and ~2 × 10^7^ CFU for *E. amylovora.*

Model setup: Apples were thoroughly washed with tap water and surface disinfected in a 0.5% sodium hypochlorite bath for 10 min. Then, they were washed thoroughly with tap water again and finally with autoclaved tap water. When dry, four holes were punched into each apple by pressing 1 mL pipette tips 10 mm deep into the apple. Prepared inoculant (10 µL) was pipetted into each hole (see paragraph above), giving four technical replicates per apple. Three biological replicates were carried out, each consisting of four technical replicates.

Glass jars (850 mL Sturzglas, Weck, Lenzburg, Switzerland) containing a Duran bottle lid (Sigma-Aldrich, St. Louis, MO, USA) serving as a 2.5 cm high pedestal were autoclaved, and 30 mL autoclaved tap water was measured into each. The inoculated apples were placed on the pedestals, keeping them out of the water. The jars’ lids were sealed with Micropore tape (3M, Maplewood, MN, USA) (Appendix A). The jars were incubated for 2.5 weeks at 26 °C. 

Bacterial quantification: After incubation, the apples were removed from their jars and photographed using fixed settings from a fixed distance. Following this, the apples were sectioned through the inoculated holes and analyzed for red fluorescence (to evaluate the amount of *E. amylovora* LMG1893-mCherry present) under a fluorescence microscope. A 218 mm^2^ area of each picture, including the entire length of the puncture, was analyzed for red signal intensity by separating the red channel, converting it to greyscale, and measuring the mean grey intensity of the picture using ImageJ (Fiji, Madison, WI, USA). The puncture sites of the water-inoculated controls were used to gain a background red fluorescence value, which was subtracted from the values for the bacterial treatments. The red fluorescence intensity values from the mixed inoculum sites were transformed by dividing by the mean intensity from sites inoculated with *E. amylovora* LMG1893-mCherry alone to give the relative intensity. 

For strains #124 and #378, *E. amylovora* LMG1893-mCherry was reisolated from the apples for CFU determination, on three biological and two technical replicates from each strain of interest. The inoculation region was homogenized with 1 mL 0.9% NaCl solution in a Bioreba extraction bag, and 200 µL of liquid was collected from the other side and diluted to a factor of 10^−6^. Dilutions were enumerated on NSLB and NSLB supplemented with 20 mg ml^−1^ gentamycin to select for *E. amylovora* LMG1893-mCherry. Furthermore, after two days at 30 °C, red colonies of *E. amylovora* LMG1893-mCherry were counted under the fluorescence microscope.

### 2.8. Tomato Leaf Model for P. infestans Pathogenicity

Two layers of paper towel were placed on the bottom of an empty 12 cm wide square plate and saturated with 7 mL autoclaved tap water. Freshly collected *Solanum lycopersicum* leaves were placed on the paper towel with the abaxial side facing up, and the petiole of each leaf was covered with an extra layer of wet paper towel. Each leaf was sprayed with ~10^8^ CFUs of the strain of interest suspended in 10 mM MgSO_4_. For the strains tested, this corresponded to approximately 1 mL of culture at OD_600_ 1. After one day at room temperature to allow colonization, a *P. infestans* plug was taken from a fresh PDA plate and placed at the point at which the leaf blade joined the petiole. The square plates were parafilmed and incubated at RT for 1 week. The leaves were placed on a light box for photographic imaging, and expansion of macerated areas was measured as distance from the point at which the leaf blade joined the petiole to the furthest point of water-soaked (translucent) tissue.

In addition to the tomato leaves that were sprayed with bacterial culture and inoculated with *P. infestans*, the following controls were prepared: leaves sprayed with bacterial culture, but no *P. infestans* plug applied; leaves sprayed with 10 mM MgSO_4_, and no plug applied; leaves sprayed with 10 mM MgSO_4_, and a sterile PDA plug applied; and leaves sprayed with 10 mM MgSO_4_, and a *P. infestans* plug applied. Three biological and five technical replicates were made for each strain.

### 2.9. Galleria mellonella Killing Assay

*G. mellonella* larvae in the final larval stage were purchased from Reptile-Food GmbH, Zürich, Switzerland. Bacterial cultures were grown overnight in NSLB at 30 °C and then subcultured 1:100 in 10 mL NSLB broth and grown under the same conditions to OD_600_ ~0.4. The bacteria were harvested, and the pellets resuspended in 10 mM MgSO_4_ (Merck). The OD_600_ was adjusted to give the desired dose. Ten µL aliquots were injected into the *G. mellonella* larvae via the hindmost proleg using a 10 µL syringe (Hamilton Company, Reno, NV, USA) with a 27 G × 7/8” needle (Terumo Agani, Shanghai International Holding Corp. GmbH, Hamburg, Germany). For the negative control larvae, 10 µL of 10 mM MgSO_4_ was injected. To avoid contamination, the injection area was disinfected before inoculation using ethanol. Ten randomly chosen larvae were used per strain tested, and each experiment was carried out in triplicate. The infected larvae were incubated in Petri dishes at 30 °C in the dark. At 20, 24, 48 and 72 h p.i., the number of dead larvae was counted. Larvae were considered dead when they did not respond to physical manipulation.

## 3. Results

This study aimed to isolate and test bacteria for use in biocontrol against fire blight, an agriculturally important disease of the *Rosacea*, most notably apple and pear trees. To increase the chances of finding strains that were able to colonize the main entry point (the blossoms) for the causative agent of fire blight (the bacterium *E. amylovora),* candidate biocontrol strains were isolated from apple blossoms. We obtained 538 isolates, which we colony-purified and tested for anti-bacterial activity against *P. carotovorum* and *E. amylovora*. In addition to this, we also tested for activity against the oomycete *Phytophthora cinnamomi* and the fungus *Fusarium solanii,* to determine whether our isolates had other traits that could make them useful for biocontrol. Nine of the isolates were found to have activity against *P. carotovorum,* although only six of these showed activity against *E. amylovora*; 161 showed activity against *P. cinnamomi*, and 117 against *Fusarium solani* (Appendix A).

Sequencing analysis of the 16S RNA-encoding gene was used to assign bacterial isolates to their genus and species (where possible), and ITS sequencing was used for fungal isolates. In total, 238 out of 538 isolates were so assigned (Appendix A). Among the isolates assigned by sequencing were the nine biocontrol candidates showing antibacterial activity. Of these, five isolates (# 93, #124, #248, #422 and #432) were members of the Operational Group *Bacillus amyloliquefaciens* (OGBa), a complex of four closely related species consisting of *B. amyloliquefaciens, Bacillus velezensis, Bacillus siamensis,* and *Bacillus nakamurai.* Due to the high 16S sequence homology between OGBa species, the *gyrB* sequence was determined for these isolates to allow species identification. The sequences of these isolates showed the highest homology to *B. velezensis*, which is known for its biocontrol properties. Isolates #378 and #379 were most likely from the species *P. agglomerans,* which is also known for its biocontrol potential but is occasionally also isolated from clinical settings. The final two antibacterial strains, #300 and #182, which inhibited the growth of *P. carotovorum* but not *E. amylovora*, were both pseudomonads, with #300 being closely related *Pseudomonas* 15A4, a typical apple blossom strain, and #182 being closely related to *Pseudomonas congelans,* a grass isolate [26,27]. Of these antibacterial isolates, only the OGBa strains and #378 showed activity against *E. amylovora.*

### 3.1. Promising Activity of #124 and #378 against E. amylovora in an Apple Model

To examine whether the in vitro anti-*E. amylovora* activity of strains #93, #124, #248, #378, #422 and #432 might translate to biocontrol activity against *E. amylovora* in planta, an apple model of infection was adapted from Klee and colleagues, 2019 [25], using an mCherry tagged *E. amylovora* strain, *E. amylovora* LMG1893-mCherry [19]. Apples inoculated with *E. amylovora* LMG1893-mCherry alone developed exocarp necrosis around the injection site, as did the apples inoculated with mixtures of a test strain and *E. amylovora* LMG1893-mCherry (Figure 1). 

In addition, persistence of *E. amylovora* LMG1893-mCherry inside the inoculation sites was evaluated by fluorescence measurement. The observed red fluorescence was significantly lower when *E. amylovora* LMG1893-mCherry was co-inoculated with OGBa #124 or *P. agglomerans* #378, compared to inoculation with *E. amylovora* LMG1893-mCherry alone, (*p* < 0.0143 and *p* < 0.0041, respectively, from a paired *t*-test) (Figure 2A). The mCherry signal was only detected at the inoculation site, and no further spread of *E. amylovora* LMG1893-mCherry through the apple tissue was observed.

To ensure that the observed decrease in fluorescence corresponded with a decrease in *E. amylovora* LMG1893-mCherry CFU, the experiment was repeated with the biocontrol strains OGBa #124 and *P. agglomerans* #378. Bacteria were re-isolated from the inoculation sites after 17 days for CFU determination. The mean CFU of *E. amylovora* LMG1893-mCherry from apples co-inoculated with test strains #124 and #378 followed the previously observed trend of being lower than when apples were inoculated with *E. amylovora* LMG1893-mCherry alone, although these differences were not significant. (Figure 2B)

### 3.2. Activity against Fungi and Oomycetes In Vitro

Many of the apple blossom isolates showed antifungal and anti-oomycete activity in our screen. We tested the isolates that showed the greatest activity against the oomycete *Phytophthora infestans*, which is the causative agent of potato late blight, and against two other fungi that were isolated from apple cores (*Diplodia seriata* and *Fusarium graminearum*) (Figure 3). 

Most of the active isolates were found to belong to the OGBa or to be fungi of the species *Metschnikowia pulcherrima*. Many strains that were found to be active against *P. cinnamomi* did not show inhibitory activity against *P. infestans*. OGBa strain #124 showed particularly promising activity in these tests, suggesting that its biocontrol activity might extend to other uses beyond the control of fire blight.

### 3.3. Biocontrol Effects of Isolated Strains against P. infestans in a Tomato Leaf Model

Strains #124, #422 and #432 showed promising results during in vitro anti-*Phytophthora* assays and were, therefore, tested on detached tomato leaves (Appendix A). None of the tested strains protected tomato leaves from developing symptoms of *P. infestans* infection.

### 3.4. Potential for Pathogenicity

In order for a strain to be useful for biocontrol, it is essential that that strain cannot cause human disease; therefore, growth at 37 °C was assessed. The seven anti-*Erwinia* candidates all showed growth at 37 °C. 

In order to further examine the strains’ pathogenic potential, they were tested in the *Galleria mellonella* larva model of infection by injecting a high bacterial CFU (~ 5 × 10^5^ bacteria per larva) (Figure 4). *G. mellonella* has an immune system consisting of cellular and humoral responses, with the former being mediated by hemocytes, and the latter by lytic enzymes, antimicrobial peptides, opsonins and melanization. Virulence in the *G. mellonella* model has been shown to generally correlate with mammalian models [28]. *B. velezensis* strain #124 showed little pathogenicity. *P. agglomerans* #378 showed dose-dependent pathogenicity, with approximately 20% of larvae surviving at 72 h p.i. with a dose of ~ 5 × 10^5^ CFU, and 50% of larvae surviving when the dose was reduced to ~ 5 × 10^4^ (Appendix A). 

We decided to sequence the genome of strain 378, to further investigate its potential as a biocontrol agent. Digital DNA–DNA hybridization was carried out on the #378 genome against that of the *P. agglomerans* ATCC 33243 type strain using the GGDC3.0 genome–genome distance calculator [29]. This confirmed that strain 378 belongs to *P. agglomerans*, (95.7% confidence interval). This result was confirmed using the TYGS online tool [29]. The strain 378 genome was subjected to antiSMASH analysis [30], which revealed the presence of putative secondary metabolite clusters. These were further investigated using BLASTX to refine their putative functions. One of the clusters identified encodes the siderophore enterobactin, one encodes the biosynthesis pathway for the carotenoid zeaxanthin, and one encodes the redox cofactor pyrroloquinoline quinone. In addition, a thiopeptide-type cluster was identified that was also annotated as a bacteriocin cluster using BAGEL [31], and that contained the *ycaO* gene, which is involved in antimicrobial activity in *E. coli* [32]. 

## 4. Discussion

Plant surfaces are considered to be inhospitable to bacterial growth, due to exposure to UV radiation, extreme fluctuations in temperature and also because of low water and nutrient availability. Cell junctions, stomata, hydathodes and nectarthodes are habitats that provide protection against direct exposure to environmental stresses [33]. A previous study of the apple blossom microbiome found the most abundant taxon present to be *Erwinia* (82% mean relative abundance), followed by *Rosenbergiella* (13% mean relative abundance) and *Pseudomonas* (2% mean relative abundance) [15]. However, it has been noted that *Pseudomonas* is sometimes the most dominant taxon in apple blossoms [34].

During our screen, we obtained 538 isolates from apple blossoms, and tested these for activity against microorganisms. While anti-oomycete and antifungal activity was relatively common, only nine isolates showed activity against *P. carotovorum,* and just seven of these were active against *E. amylovora*. We were able to sequence type 238 isolates from our collection of 538, and all were typical of the known apple blossom microbiome, with a high incidence of *Pseudomonas* and *Bacillus* species (Appendix A) [15].

We tested the strains showing the highest antifungal and anti-oomycete activity against two other fungi and one further oomycete. This revealed *B. velezensis* strain #124 as the most active strain against these organisms in vitro. Other OGBa strains are currently approved for the control of fungal diseases of plants, in the EU namely *Bacillus amyloliquefaciens* AH2, IT45, FZB24 and D747, and *Bacillus velezensis* QST713, MBI 600 and RTI301. These are the active ingredients in fungicides such as SERENADE Max, TAEGRO, Kodiak and Double Nickel 55 [35,36]. In addition, we found several different *M. pulcherrima* yeast strains (#118, #195, #196, #197, #332, #482 and #483) that were highly active against *P. cinnamomi* in vitro, and which might be interesting candidates for the development of a biocontrol agent.

This study found two promising candidates for biocontrol against fire blight, OGBa member *B. velezensis* #124 and *P. agglomerans* #378. While neither of these strains prevented necrosis of the apple exocarp around the inoculation site when co-inoculated with *E. amylovora*, such necrosis is an instant response of the apple upon detection of a foreign organism, known as the hypersensitive response (HSR). Such lesions result from programmed cell death, designed to prevent the pathogen from spreading to healthy tissue [37]. Strains *B. velezensis* #124 and *P. agglomerans* #378 significantly reduced detected mCherry fluorescence around the inoculation site in the apple model. This trend was also observed when CFUs were counted, although the difference was not significant. In conclusion, our data suggest that *B. velezensis* #124 and *P. agglomerans* #378 have good potential as biocontrol agents. 

The OGBa is part of the *B. subtilis* complex and currently consists of *B. amyloliquefaciens*, *Bacillus velezensis, Bacillus siamensis* and *Bacillus nakamurai* [36,38]. *B. velezensis* strains such as the registered biocontrol agents FZB24 and FL50S are known to be active against *E. amylovora*. Their mode of action relies on the production of antimicrobial polyketides such as difficidin, bacillaene and oxydifficidin [10,39]. The activity of *B. velezensis* isolate #124 was observed as a halo on agar plates, and is, therefore, likely to be mediated by diffusible antimicrobial agents, similar to the currently registered biocontrol strains.

It is likely that the antibacterial activity of isolate #378 that we observed in vitro is the result of a bacteriocin that was able to diffuse into the agar plates, perhaps specified by the thiopeptide type cluster identified by antiSMASH and BAGEL [30,31]. Anti-*Erwinia* activity of *P. agglomerans*, formerly classified as *Erwinia herbicola,* has been previously reported. In contrast to our *P. agglomerans* isolate, the reported anti-*Erwinia* activity of *P. agglomerans* NY60 required cell–cell contact. The activity of *P. agglomerans* Eh1087 was mediated by phenazines, which is unlikely to be the case with strain #378, as no phenazine biosynthetic cluster was detected within its genome. Therefore, strain #378 could be an interesting candidate for further investigation. 

*P. agglomerans* is a ubiquitous environmental bacterium, found in plants and feces. It is not a primary pathogen, but some strains can cause opportunistic infections in humans, mostly either as a result of skin breakage by plant material (such as a thorn wound) or in the hospital setting [28]. *P. agglomerans* infections generally respond well to antibiotic treatment and, therefore, tend to have favorable outcomes [30]. *P. agglomerans* strain P10C (Blossom Bless) is registered for agricultural use in New Zealand https://www.epa.govt.nz/database-search/hsno-application-register/view/HSR05024 accessed on 10 November 2022). Two strains of *P. agglomerans* were previously registered for use in the USA and Canada, but their registration has since been canceled (BlightBan and Bloomtime, *P. agglomerans* strains C9-1 and E325, respectively, http://npic.orst.edu/NPRO/ accessed on 10 November 2022). Due to the difficulties in distinguishing biocontrol from clinical strains, *P. agglomerans* cannot currently be used for agricultural applications within the EU [40].

Future work will be directed at testing our two candidates, *P. agglomerans* #378 and *B. velezensis* #124, in further models, such as detached apple blossoms and pear fruitlets, and following that in trees. It is well-known that in vitro antagonism does not always translate to activity in the field [4]. However, it is promising that the results observed in vitro for strains #378 and #124 have translated successfully to antagonism in the detached apple model, making these strains worthy of continued investigation.

Our results showcase the potentially protective microorganisms living in the apple blossom niche. We found two candidate biocontrol strains that are members of species that are currently in use for biocontrol (although use is geographically restricted for *P. agglomerans*). Since they were originally isolated from apple blossoms, and given that related strains are currently used for biocontrol of fire blight, it is likely that they can persist at the most common entry point for *E. amylovora*. Since these strains were isolated from apple blossoms, they are likely to already be part of the human diet; indeed, investigations of the apple fruit microbiome have previously found members of the genera *Bacillus* and *Pantoea* within apple fruits, with *Pantoea* members becoming more prevalent upon post-harvest storage [41,42]. This suggests that they might be safe for consumption and promising for future investigation.

## Figures and Tables

**Figure 1 microorganisms-10-02480-f001:**
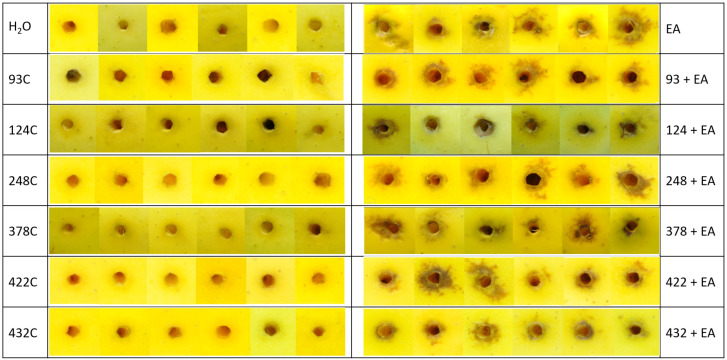
Appearance of peal area around the injection site 2.5 weeks post inoculation. Left: controls with water inoculation or biocontrol test strains alone. Right: *E. amylovora* LMG1893-mCherry (EA) mixed with the bacterial strains for inoculation. Strains tested were #93, #124, #248, #378, #422, and #432. Six biological replicates were performed.

**Figure 2 microorganisms-10-02480-f002:**
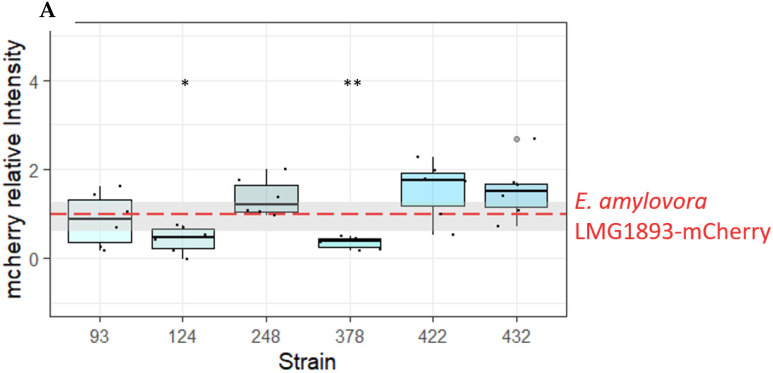
Isolates active against *E. amylovora* in an apple model of infection. (**A**) Strains OGBa #124 and *P. agglomerans* #378 antagonized *E. amylovora* LMG1893-mCherry in the apple model of infection. The intensity of mCherry in the apple model was measured for *E. amylovora* LMG1893-mCherry mixed with strains #93, #124, #248, #378, #422, or #432 and compared to mCherry intensity for the *E. amylovora* LMG1893-mCherry single strain inoculant (red line). **, *p * < 0.01; * *p* < 0.05 very significant compared to *E. amylovora* LMG1893-mCherry single inoculant with *p* < 0.001(paired *t*-test). Mean and interquartile range are indicated by bold line and box, respectively. (**B**) Strains OGBa #124 and *P. agglomerans* #378 reduced *E. amylovora* CFUs after 2.5 weeks of incubation. Bacteria were re-isolated from apple model inoculation sites after 17 days at 26 °C, and CFUs of *E. amylovora* LMG1893-mCherry were counted for apples inoculated with *E. amylovora * LMG1893-mCherry only (EA) and for the mixtures with strains #124 or #378. Mean and interquartile range are indicated by bold line and box, respectively.

**Figure 3 microorganisms-10-02480-f003:**
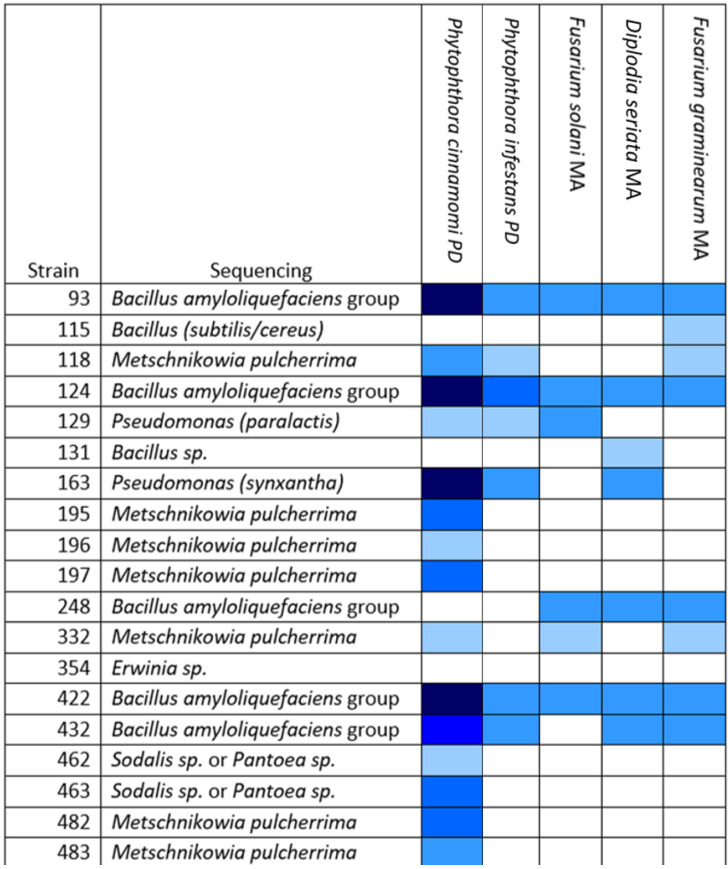
Several isolates show activity against a range of oomycetes and fungi. Test strains (numbered left) in dual culture plate assay against pathogens labeled across the top of the table. PD, potato dextrose agar; MA, malt extract agar. Zones of inhibition (ZOIs) between the test bacterium/fungus and pathogen were measured when the leading edge of the pathogen approached the edge of the control plate (without bacterial antagonists). The depth of the blue color in the matrix shown represents the size of the ZOI: • 1–5 mm, • 5.1–10 mm, • 10.1–15 mm, • 15.1–20 mm, • 20.1–25 mm. White: no ZOI.

**Figure 4 microorganisms-10-02480-f004:**
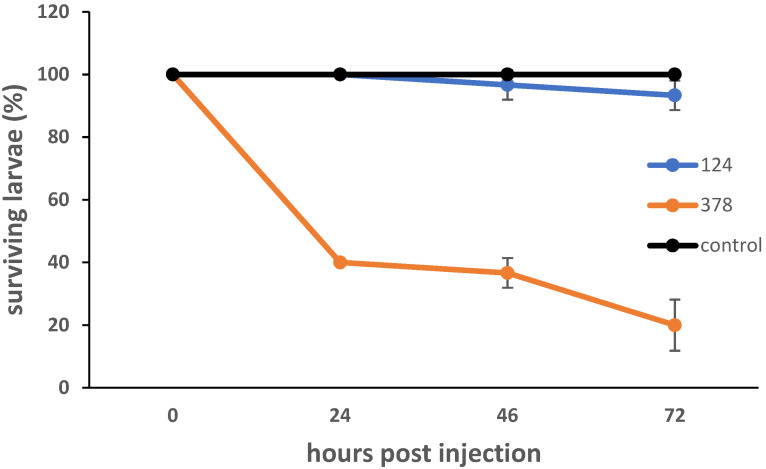
Survival curves for *G. mellonella* larvae infected with strains *B. velezensis* #124 and *P. agglomerans* #378. Larvae were injected with approximately 5 × 10^5^ bacteria and incubated at 30 °C in the dark. Live and dead larvae were counted at 24, 48 and 72 h p.i. Curves represent the mean of three separate experiments. Error bars represent the standard deviation of the data.

**Table 1 microorganisms-10-02480-t001:** Bacteria, fungi and oomycetes used as test pathogens in this study.

Strain	Description	Reference or Source
*Erwinia amylovora* LMG1893	wild type, *Pyrus communis* isolate	Vantomme et al., 1982 [18]
*Erwinia amylovora* LMG1893-mCherry	LMG1893 tagged by insertion of mini Tn7(Gm^r^)PA1/04/03-mCherry	Putschert-Montenegro et al., 2021 [19]
*Erwinia carotovora* LMG2404	wild type, isolated from *Solanum tuberosum*	Hauben et al., 1998 [20]
*Fusarium solani*	wild type	Federal Institute of Technology, Zurich, Switzerland
*Fusarium graminearum*	Isolated from an apple, wild type	This study
*Diplodia seriata*	Isolated from a rotten apple, wild type	This study
*Phytophthora cinnamomi* Rands P 563.12	wild type	Silva Tarouca Research Institute for Landscape and Ornamental Gardening (RILOG), Průhonice, Czech Republic
*Phytophthora infestans* (Mont.) de Bary Rec01	wild type	Agroscope, Zürich-Reckenholz

**Table 2 microorganisms-10-02480-t002:** Primers used in this study.

Name	Sequence	Description	T_A_ ^1^	Reference
PRK341F	CCTACGGGRBGCASCAG	For 16S rRNA sequencing	55 °C	Yu et al., 2005 [21]
PRK806R	GGACTACYVGGGTATCTAAT			
UP-1	GAAGTCATCATGACCGTTCTGCAYGCNGGNGGNAARTTYGA	For gyrase B (*gyrB*) sequencing	60 °C	Yamamoto and Harayama, 1995; [22] Wang et al., 2007 [23]
UP-2r	AGCAGGGTACGGATGTGCGAGCCRTCNACRTCNGCRTCNGTCAT			
BS-Fv	GAAGGCGGNACNCAYGARG	Internal *gyrB* primers for differentiation of OGBa species		Wang et al., 2007 [23]
BS-Rv	CTTCRTGNGTNCCGCCTTC			
ITS4-5F	TCCTCCGCTTATTGATATGC	For internal transcribed spacer (ITS) sequencing	55 °C	White et al., 1990 [24]
ITS4-5R	GGAAGTAAAAGTCGTAACAAGG			

^1^ annealing temperature.

## Data Availability

The genome of *P. agglomerans* #378 is available at NCBI under the accession numbers CP113085–CP113087.

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
