# Peer review of "Investigating the Biocontrol Potential of the Natural Microbiota of the Apple Blossom"

_microorganisms, 2022, doi:10.3390/microorganisms10122480_

Round 1
Reviewer 1 Report
The article titled “Investigating the biocontrol potential of the natural bacterial flora of the apple blossom” revealed the bacterial flora of the apple blossom in combatting plant fire-blight. They isolated various strains and tested for activity against P. carotovorum, E. amylovora, and oomycete and fungal pathogens. They reported two isolates that showed activity against all tested pathogens and that were capable to antagonise E. amylovora in an apple model of infection. This is a well-written article and I anticipate that the manuscript should be of great interest to the researchers working on biocontrol of bacterial flora. Before recommending this article for publication, there are some shortcomings for that should be resolve.
Overall, the study is well-designed and presented in a good way. However, many sentences include repetitive words and statements that should be resolved.
The authors elaborated the abstract in a good way. However, some points need to be addressed to improve this section.
Organize the abstract in a structured way and add some noticeable results.
Line 19, what were the names of three different fungi?
Add two or three more keywords
Line 28-30; Revise the statement and retain the word production or yields, not both and also add a relevant reference.
Add one more sentence at the end of the first paragraph for coherence to the following (second) para.
The introduction part is well written but several statements must be cited.
Can you add some details on the legislation and biocontrol policies, globally?
You developed the methods for antibacterial, Anti-Phytophthora, and antifungal assays or followed some other works. If so please cite accordingly.
Do not start a sentence from numbers such as Lines 113, 123, 149, and others.
Results are presented and elaborated in a proper way.
Literature cited and discussed in a proper way (well written and presented).
References: Appropriate and adequate references to related works are covered sufficiently in the list. Incorporate some latest references.
Reviewer 2 Report
In the comments below are some minor comments and suggestions for improving the Manuscript:
Page 1, Line 18: Add how many isolates were genetically determined.
Page 1, Line 25: Add another keyword: “Apple blossom bacteria”.
Page 1, Lines 41-46: Include some more references for this section.
Page 2, Line 84: Explain why were the strains incubated at 37°C. Was it due to pathogenicity test?
Page 3, Line 95: Explain were these strains used as pathogens in biocontrol tests of isolated bacteria.
Table 1: Minimise the font size in the Tables 1 and 2.
Page5, Line 119: Add approximate temperature of agar, it should be not above 45°C.
Page 6, Line 149: Explain the criteria for the selection of bacteria to be sequenced. Did you construct concatenated sequence based on 16s rRNA and gyrB sequence. The construction of concatenated phylogenetic tree for important isolates should be provided as figure in the results section.
Page 6, Line 158: Replace yeasts with fungi.
Page 6, Line 169: The section Genomic DNA preparation for sequencing could be merged with Section 2.4, as an independent paragraph.
Page 6, Lines 183-188: Explain here how the apples were infected with E. amylovora.
Page 8, Lines 260-271: Is this based only on 16s rRNA sequences? Results of gyrB sequencing should also be mentioned.
Page 8, Lines 275-279: Move this to Materials and Methods section.
Discussion section: Adding some more references regarding the useful bacteria isolated from apple blossoms (if any), or bacteria with biocontrol potential isolated from other parts of apples (including rhizosphere) would be useful.
Reviewer 3 Report
Some remarks:
Line 2: the title should be changed, the term flora is used sometimes, but it is incorrect
L. 9: as above
L. 12: whether strains or isolates were tested?
L. 16: all classes ???
L. 22: they can persist at the most common entry points???
L. 23: human diet ??? – too bold a statement, especially in the context of P. agglomerans
L. 25: key words are for fire blight only, is that enough?
L. 35-36: it is worth adding what about the effectiveness of biopreparations, whether they are competitive in all conditions?
L. 42: not only the flowers are the site of infection, add other affected organs of the aerial part of plants
L. 52-58: add in which phenological stages the biological method can be recommended and its possible use in the integrated plant protection system, moreover, what are the limitations of the method, especially in the context of environmental conditions
L. 59: bacterial flora?
L. 61: strains or isolates?
L. 62: the results of in vitro tests may indicate the potential of the tested bacteria, but very often they do not translate into natural conditions
L. 66: strains is not the same as isolates and the terms are not always interchangeable
L. 82-97: what is the importance of the selection of bacterial isolates from different taxa on the basis of antagonism on one medium (in the literature there are information about obtaining different results on different media); in addition, the absence of antagonism on the medium does not mean that the isolate will not be protective in the orchard
L. 111: should be P.cinnamoni
L. 176: whether the method published by Klee... using apples is sufficiently reliable in the selection of isolates in terms of their practical application, what about other methods, e.g. using apple and pear flowers or pear fruitlets?
L. 180: specify the exact concentration of the inoculum?
L. 335-343: please give a broader comment on the pathogenicity of the tested isolates (124 and 378) towards Galleria mellonella larvae from the point of view of their practical use
L. 433: detection of isolated bacteria on flowers or other organs does not mean that they will multiply there and show protective abilities
w. 477: the results of the conducted research show the potential of the selected bacteria, but further research is required to use them in practice
Reviewer 4 Report
My comments and suggestions on each part of the manuscript are listed below.
Abstract
The Abstract could be more focused on the most essential outcomes of the experimental work. It says that samples of apple blossoms were collected from several sites differ on application or not of conventional pesticides. As this information is not directly connected with the main results that are presented in the manuscript, it is better to mention it only in Material and methods section or you have to make some relationship between usage of additional agents and obtained results for microbial diversity.
Materials and methods
This section is well structured and contains all necessary information.
Row 242 - it is better to use “insects” instead of “animals”
Results
The results are presented in details and the figures are informative. However, there are some sentences that are part of Material and methods already and they are redundant in the Result section.
Starting from the first paragraph, only sentences like “We obtained 538 isolates, which we colony purified and tested for anti-bacterial activity against P. carotovorum and E. amylovora…. Nine of the isolates were found to have activity against P. carotovorum, although only six of these showed activity against E. amylovora, 161 showed activity against P. cinnamomi, and 117 against Fusarium solani” have to remain in this part of the manuscript. Authors have to remove “The blossoms were homogenized, and bacterial isolates selected on 252 three media (LB, M9 and PCAT)” and other methodological descriptions and further in the text.
Rows 275-279 – these sentences are not a result; the explanation is a part of Materials and methods
Rows 415-416 - this is not a result
My suggestion to the authors is to reorganize some parts of results section. It would be better if “Potential for pathogenicity” remains in the end of the Results. In this way it will correspond to the order of methods and will make a link to Discussion.
Row 338 has to be a new paragraph.
Rows 360-375 – this part is for Discussion, not a direct result of experiments presented in the manuscript
Discussion is informative and well structured.
There are some small details that have to be improved in the revised version:
Rows 456-457 – do not use “previously” twice in one sentence
Rows – 467-468 – “other” is duplicated
References
Check all references for technical inaccuracies. There are years of some publications that are not in bold (ref. 4, 14, 15, etc.) and some references are without pages or volume of the journal (ref. 31, 32).
